# Dietary Patterns, Metabolomic Profile, and Nutritype Signatures Associated with Type 2 Diabetes in Women with Postgestational Diabetes Mellitus: MyNutritype Study Protocol

**DOI:** 10.3390/metabo12090843

**Published:** 2022-09-07

**Authors:** Farah Yasmin Hasbullah, Barakatun-Nisak Mohd Yusof, Rohana Abdul Ghani, Zulfitri ’Azuan Mat Daud, Geeta Appannah, Faridah Abas, Nurul Husna Shafie, Hannah Izzati Mohamed Khir, Helen R. Murphy

**Affiliations:** 1Department of Dietetics, Faculty of Medicine and Health Sciences, Universiti Putra Malaysia, Serdang 43400, Selangor, Malaysia; 2Research Centre of Excellence for Nutrition and Non-Communicable Diseases, Faculty of Medicine and Health Sciences, Universiti Putra Malaysia, Serdang 43400, Selangor, Malaysia; 3Institute for Social Science Studies, Putra Infoport, Universiti Putra Malaysia, Serdang 43400, Selangor, Malaysia; 4Department of Internal Medicine, Faculty of Medicine, Universiti Teknologi MARA, Sungai Buloh 47000, Selangor, Malaysia; 5Department of Nutrition, Faculty of Medicine and Health Sciences, Universiti Putra Malaysia, Serdang 43400, Selangor, Malaysia; 6Department of Food Science, Faculty of Food Science and Technology, Universiti Putra Malaysia, Serdang 43400, Selangor, Malaysia; 7Norwich Medical School, University of East Anglia, Norwich NR4 7TJ, UK

**Keywords:** type 2 diabetes, gestational diabetes mellitus, cardiometabolic, postpartum, dietary patterns, metabolomics, nutritype

## Abstract

Women with previous gestational diabetes mellitus (post-GDM) have an increased risk of cardiometabolic diseases including type 2 diabetes (T2D). Current diabetes screening is based on the oral glucose tolerance test without nutritional assessments, even though unhealthy dietary patterns were found to expedite disease progression in women post-GDM. While a healthful dietary pattern reduces T2D risk, limited data support a dietary pattern tailored to the Asian population, especially in the Malaysian context. Metabolomic profiles associated with dietary patterns in this population are also lacking. The proposed study aims to investigate both components of dietary patterns and metabolomic profile, known as nutritype signatures, and their association with T2D in women post-GDM. The comparative cross-sectional study will involve a minimum of 126 Malaysian women post-GDM aged 18–49 years. Dietary patterns will be analysed using principal component analysis. Plasma and urinary metabolites will be quantified using one-dimensional proton nuclear magnetic resonance (^1^H NMR) spectroscopy. The aim of the study is identifying the nutritype signatures associated with T2D. The findings will support the development of early prevention measures against T2D in women post-GDM.

## 1. Introduction

Type 2 diabetes (T2D) has reached epidemic proportions in Asia, with the Western Pacific Region contributing to 38% of T2D cases worldwide [1]. Malaysia is the fifth leading country in the Western Pacific Region with the highest prevalence of T2D [1]. The prevalence of T2D in Malaysia has increased by almost 60% over a 10-year period, partly due to a rising prevalence of overweight and obesity, physical inactivity, and dietary practices that include a low intake of fruits and vegetables and high intake of sugar-sweetened beverages [2].

Women previously diagnosed with gestational diabetes mellitus (post-GDM) had a nearly 10-fold relative risk of T2D compared with those with normoglycemic pregnancies [3]. Hence, early identification and intervention are critical to preventing T2D in this high-risk group. Risk factors of T2D in women post-GDM include high prepregnancy body mass index (BMI), sedentary behaviour, and an unhealthy dietary pattern low in carbohydrates but high in animal protein and fat [4,5,6]. Current recommendations support diabetes screening at 6 weeks postpartum via oral glucose tolerance test (OGTT) and/or HbA1c after GDM [7]. However, diabetes screening conventionally does not include nutritional assessments, even though an unhealthy diet is highly relevant in expediting disease progression in women post-GDM [6].

Regarding dietary patterns, adherence to healthful dietary patterns such as the alternate Mediterranean Diet (aMED) was associated with a lower risk of T2D in women post-GDM [8]. The Mediterranean diet is widely acknowledged as a healthy dietary pattern, characterised by high intakes of vegetables, fruits, nuts, grains, legumes, fish, and olive oil [9]. Malaysian adults consume a diet high in fish and green leafy vegetables, yet simultaneously high in rice, sugar, animal protein, and fat [10,11], showing a dissimilarity with dietary patterns commonly practiced in the Western population. Limited research has been conducted on the association between Malaysian-tailored dietary patterns and T2D risk in women post-GDM.

As the nutritional assessments have shifted the focus on dietary patterns, we need to identify biomarkers of dietary patterns via the metabolomics approach. Metabolomics is a tool to measure the full profiles of low-molecular-weight metabolites in biofluids such as blood plasma and serum and urine [12]. Metabolites that can predict the transition from GDM to T2D at postpartum have been identified in various studies, including branched-chain amino acids (BCAAs), hexoses, fatty acids, sphingomyelins, and carnitines [13,14,15,16,17,18]. Nevertheless, most of these studies were limited to Western populations and did not include Asian ethnic groups.

The application of metabolomic techniques in nutrition research has led to the emerging concept of nutritype [19]. Nutritype refers to the expression of overall dietary intake in metabolites and can be used to classify individuals into a specific dietary pattern based on their metabolomic profile [19]. In other words, dietary patterns are analysed together with metabolomic profiles to identify the nutritype signatures of the study population [19].

While the roles of dietary patterns and metabolomics in determining T2D risk are significant, these two parameters have usually been separately studied. Current evidence on the combination of dietary patterns and metabolomic profile (nutritype signatures) in women post-GDM has been inconclusive, with varying results being reported [14,15,16,17]. All of the four studies only focused on specific components of dietary patterns and their metabolomic markers, such as cooking fat and oil [14], dietary fatty acids [15], glycaemic load [16], and dietary BCAAs [17]. None of the studies observed dietary patterns in their entirety. The scant evidence indicates a gap in the knowledge and a pressing need for further studies on nutritype signatures to capture the exposure to T2D in women post-GDM.

This paper describes the protocol of the Malaysian Nutritype (MyNutritype) study, including the methodology, research workflow, outcome measures, and statistical analysis. The study aims to investigate the association between dietary patterns, metabolomic profile, and T2D in women post-GDM.

## 2. Methods

This section describes the finalised study design, study population characteristics, and research workflow of the proposed study.

### 2.1. Design

This is a comparative cross-sectional study involving women post-GDM, named the Malaysian Nutritype (MyNutritype) study. The primary outcome is the nutritype signatures associated with T2D in Malaysian women post-GDM. The secondary outcomes of the study include the comparisons of anthropometric and clinical measurements, biochemical profile, energy and nutrient intake, and lifestyle practices between the three groups of women post-GDM (NGT, prediabetes, and T2D).

The aims of this study are:(a)To determine dietary patterns associated with T2D and prediabetes;(b)To determine the metabolomic profile associated with T2D and prediabetes using one-dimensional proton nuclear magnetic resonance (^1^H NMR) spectroscopy;(c)To identify nutritype signatures (association between dietary patterns and metabolomic profile);(d)To determine the nutritype signatures associated with T2D in women post-GDM;(e)To compare other parameters in women post-GDM with normal glucose tolerance (NGT), pre-diabetes and T2D, which include:i.Anthropometric and clinical measurements;ii.Biochemical profile;iii.Energy and nutrient intake;iv.Lifestyle practices.

### 2.2. Study Setting

The study will be conducted at two sites: Universiti Putra Malaysia (UPM) and Klinik Kesihatan Seri Kembangan. UPM is a tertiary institution and the overall responsible site for this study. The recruitment site at UPM is the Faculty of Medicine and Health Sciences. Klinik Kesihatan Seri Kembangan is a government health clinic that includes a medical outpatient clinic and a maternal and child health clinic. Both sites are located in the Petaling District of Selangor, Malaysia. ^1^H NMR spectroscopy and metabolomic analysis will be performed at the Institute of Bioscience, UPM.

### 2.3. Study Population

The study will include Malaysian women aged 18–49 years with a previous diagnosis of GDM. Exclusion criteria will be the following: pregnant women, recent hospitalisation within the last 6 weeks, and previously diagnosed medical problems (such as type 1 or type 2 diabetes, cancer, and liver or renal disorders). The sample size will be estimated using the mean difference formula [20]. Based on a study that compared dietary intake and metabolomic profile in NGT, prediabetes and metabolomic profile in women post-GDM in Gothenburg, Sweden [14], considering a 95% confidence level and 80% power, a total of 99 subjects will be needed for the study. To account for nonresponse or refusal to participate, an additional 20% of subjects will be required, leading to a minimum of 126 subjects.

### 2.4. Recruitment

Women who fit the selection criteria will be invited to join the study. Subjects will then be selected using systematic random sampling. Informed written consent will be obtained at enrolment. Subjects will be briefed on the benefits and potential risks of joining the study.

Subject recruitment commenced in March 2021, and enrollment is expected to continue through October 2022. The research workflow is shown in Figure 1.

### 2.5. Measures

The outcomes of the study are mainly the anthropometric data, biochemical parameters, and energy as well as nutrient intakes. The primary variables of interest are dietary patterns and metabolomic profiles.

#### 2.5.1. Diagnosis of Type 2 Diabetes and Prediabetes

After an overnight fast of 8–12 h, subjects will undergo a 75 g, 2 h OGTT. Blood and urine samples will be taken for HbA1c, insulin, lipid profile, and metabolomic analysis. Subjects will be grouped into either NGT, prediabetes or T2D based on their OGTT and HbA1c results, as advised by the study endocrinologist. Our diagnostic criteria for T2D and prediabetes are based on the Malaysian Clinical Practice Guidelines [7] (Table 1).

#### 2.5.2. Sociodemographic Characteristics and Obstetric History

The study will collect sociodemographic data on age, ethnicity, education level, marital status, occupation, household income, and household size using a structured questionnaire. A family history of diabetes will also be assessed. Information on obstetric history will be obtained from the antenatal records. This will include gravidity, parity, pre-pregnancy weight, gestational weight gain, postpartum weight retention, number and timing of GDM diagnosis, duration lapse since GDM diagnosis, treatment during GDM, delivery method, birth outcomes, infant birth weight, breastfeeding duration, and family planning.

#### 2.5.3. Dietary Patterns

The study dietitian will administer a semiquantitative food frequency questionnaire (FFQ) adapted from the nationwide Malaysian Adult Nutrition Survey, to determine food consumption in the past month [21]. It contains 14 food groups: grain and cereal products; fast food; meat and poultry; fish and seafood; eggs; legumes, seeds and nuts; milk and dairy products; vegetables; fruits; beverages; alcoholic drinks; confectionaries; bread spread; and condiments. The portion size of the food items will be according to the Food Portion Sizes of Malaysian Foods Album [22]. The amount consumed food will then be converted into grams per day for each food item [10].

Dietary patterns will be derived based on PCA. Data suitability for factor analysis will be assessed prior to performing PCA [23,24]. Food items will be classified based on similarities in nutrient profiles and previous local studies [24,25,26]. The factor scores will be orthogonally rotated by varimax transformation to increase loading differences for easier interpretability [24,27]. Eigenvalue cut-off > 1.5 will determine the number of factors [28]. Food items with factor loadings < ±0.3 will be removed from further analysis [24,28]. A high factor score, whether positive or negative, indicates a high intake of that food group in dietary pattern; and vice versa [23,27].

#### 2.5.4. Energy and Nutrient Intake

The study dietitian will administer 3-day 24 h dietary recalls. Subjects will be asked about the food and beverages consumed in the past 24 h. Other details will include the meal time, type of food and beverage, cooking method, and quantity or portion size. On the day of the visit, a 24 h dietary recall will be obtained. The research dietitian will later interview the subjects via phone calls to obtain another 2 days of 24 h dietary recalls. The 3 days of 24 h dietary recalls will be on nonconsecutive days (2 weekdays and 1 weekend).

Energy and nutrient intake calculations will be performed by the Nutritionist Pro software (V.5.1.0, Axxya Systems, Redmond, WA, USA). The serving size of the food and data on energy and nutrients will be obtained from the Nutrient Composition of Malaysian Foods database on the Nutritionist Pro software. If the food or beverage is not listed in the Malaysian food database, data will be obtained from the Singapore Food Composition database [29] and the U.S. Department of Agriculture database [30]. Recipes for local cuisine or mixed dishes will be created if they are unavailable on all databases. The average of 3-day dietary recalls will be obtained to determine the mean intake of energy and nutrients, including carbohydrates, protein, fat, fibre, vitamins, and minerals.

#### 2.5.5. Lifestyle Practices

The lifestyle practices include dietary supplements, sleep duration, stress level, physical activity and smoking habit. Subjects’ frequency and dosage of dietary supplements will be obtained using a questionnaire adapted from the Malaysian Adult Nutrition Survey [21]. Sleep duration will be reported as the total hours of sleep in an average 24 h period over the past month [31]. The study will assess subjects’ stress levels using the 10-item Perceived Stress Scale (PSS-10) [32], which was validated in Malaysian T2D patients [33]. Physical activity level will be assessed using the International Physical Activity Questionnaire-Short Form (IPAQ-SF) [34]. We will use the Global Adult Tobacco Survey to assess smoking habits and second-hand smoking exposure (at home, work, public transport, and closed public areas) among subjects [35].

#### 2.5.6. Anthropometric and Clinical Measurements

Height will be measured using a stadiometer (SECA model 206, Vogel & Halke GmbH & Co., Hamburg, Germany). Subjects will be asked to remove their outer attire and shoes and empty their pockets, then stand with their back to the stadiometer while looking straight ahead. Measurements will be recorded to the nearest 0.1 cm. A body composition monitor will be used to measure the body weight and body fat percentage (Tanita Health Equipment Ltd., Tokyo, Japan). BMI will be calculated and categorised according to the World Health Organization (WHO) classification [36]. We will measure waist and hip circumferences using a measuring tape (SECA model 203, Vogel & Halke GmbH & Co., Hamburg, Germany), and the waist-to-hip ratio will be categorised according to WHO [37]. A hand dynamometer will be used to measure grip strength (JLW Instruments, Chicago, IL, USA), and the mean grip strength will be compared with the consolidated grip strength reference values [38]. Blood pressure will be measured using a blood pressure monitor (OMRON Corporation, Kyoto, Japan) and categorised according to the Adult Treatment Panel (ATP III) guidelines [39].

#### 2.5.7. Biochemical Profile

Subjects will be asked to fast overnight for 8–12 h prior to blood collection. Fasting venous blood samples will be drawn by a trained phlebotomist. Blood samples for conventional biochemical parameters (glucose, HbA1c, insulin, and lipid profile) will be transported to an established commercial laboratory for analysis (Clinipath Sdn. Bhd., Bangsar, Malaysia). Venous blood will be drawn at 2 h following the administration of a 75 g glucose solution drink to obtain the postprandial glucose level.

#### 2.5.8. Metabolomic Profile

NMR sample preparation for plasma and urine will be performed according to published protocols [40,41]. About 2 mL of the fasting blood will be collected into lithium heparin tubes, centrifuged at 1500× *g* for 20 min, divided into aliquots, frozen in a liquid nitrogen tank, and stored at −80 °C until further analysis. Subjects will also have to collect mid-stream urine into a urine container. Sodium azide 0.1% will be added to the urine sample to prevent microbial growth. The urine samples will be centrifuged at 1500× *g* for 10 min, divided into aliquots, and stored at −80 °C until further analysis.

Upon thawing, 500 μL of plasma will be vortexed for 1 min, then centrifuged for 2 min at 5000× *g* to remove solid debris. We will transfer 400 μL of plasma supernatant into a 0.5 mL ultra-centrifugal filter and centrifuged at 6000× *g* for 30 min to remove macromolecules, including lipids and proteins. We will mix 200 μL of plasma filtrates with 400 μL of phosphate buffer solution (potassium dihydrogen phosphate Kh_2_Po_4_]) prepared in deuterium oxide (D_2_O) containing 0.1% trimethylsilyl propanoic acid (TSP) and 0.1% imidazole). A total of 600 μL of prepared samples will be transferred to 5 mm NMR tubes. For urine, 400 μL of thawed sample will be added to 200 μL of phosphate buffer solution, then centrifuged at 4800× *g* for 5 min before being transferred to NMR tubes.

The nontargeted metabolomic analysis will be performed using a ^1^H-NMR platform. ^1^H-NMR spectra will be acquired using a 500 MHz NMR spectrometer (Varian Inc., Palo Alto, CA, USA). Spectral data will be preprocessed using Chenomx NMR suite software version 9.0 (Chenomx Inc., Edmonton, AB, Canada) by Fourier transformation, auto-phasing, and auto-baseline correction. Spectral data will be divided into 1000 equal bins, digitised into a table of common integral, and exported as a text file for multivariate data analysis.

Multivariate data analysis of the processed spectra will be performed using SIMCA-P+ version 15.0 (Sartorius Stedim, Umeå, Sweden) by applying various statistical algorithms including principal component analysis (PCA), partial least square discriminant analysis (PLS-DA), and orthogonal projections to latent structures discriminant analysis (OPLS-DA) to determine differences in metabolites between the three groups. Based on the multivariate data analysis results, parts of the spectra (ppm) responsible for the differences between groups will be quantified using Chenomx software by matching the corresponding peak with the metabolite database. PCA score and loading plots will be drawn in SIMCA-P+ software to validate the data obtained from Chenomx software. Pathway identification will be based on the Kyoto Encyclopedia of Genes and Genomes (KEGG) (Kanehisa Laboratories, Kyoto, Japan).

#### 2.5.9. Nutritype Signatures

Nutritype signature models will be developed based on the three-step chemometric strategy for nutrimetabonomics [42,43]. In step 1, dietary patterns, as assessed by PCA, will be used to identify the factor scores as described in Section 2.5.3. The factor scores will then be designated as “loadings (VFFQ)”. In step 2, factor scores will be coded to create dummy variables (UFFQ). Positive scores will be coded as class number one, whereas negative scores will be coded as class number two. In step 3, the coded factor scores of each dietary pattern will be regressed against the ^1^H NMR metabolite profile to identify the metabolic phenotypes (“loadings, VNMR”) associated with these dietary patterns using PLS-DA to maximise the separation between T2D and non-T2D groups. The corresponding loading plots from the PLS-DA analyses will determine the significant metabolites we will use to develop nutritype signature models [25]. Chemometric analysis will be performed to produce nutritype signature models using SIMCA-P 15.0+ software (Sartorius Stedim, Umeå, Sweden).

### 2.6. Governance and Ethics

Ethical approval for the study was obtained from the Medical Research Ethics Committee (MREC) of Ministry of Health, Malaysia (NMRR-19-3482-50546) and Research Committee of Universiti Putra Malaysia (JKEUPM) (JKEUPM-2019-464).

The research team comprises a multidisciplinary team, which includes the principal investigator (PI), an endocrinologist, a dietitian, a biochemist, and a Ph.D. student. The PI will develop the study protocol, verify progress reports to the funder, oversee the study progress, and ensure the scientific accuracy of study results. The endocrinologist will oversee the OGTT process and stratify subjects into NGT, prediabetes, or T2D groups based on the OGTT and HbA1c results. The dietitian will administer the food frequency questionnaire (FFQ) and the 3-day dietary recalls; take anthropometric measurements; and calculate the energy and nutrient intakes of the subjects. The biochemist will be responsible for offsetting up and storing metabolomic reagents and buffers; plasma and urine sample storage; liaising with the metabolomic science officer regarding ^1^H NMR sample run; preprocessing spectral data on Chenomx NMR suite software; metabolite profiling; and metabolomic pathway identification. Finally, the Ph.D. student will be responsible for the day-to-day management of the study; procurement of equipment and chemicals; training of enumerators; subject recruitment and enrolment at the study sites; liaising with commercial laboratory regarding blood tests; and centrifugation of plasma and urinary samples for metabolomic analysis. The Ph.D. student will also be responsible for communicating with the medical doctors, nurses, phlebotomists, and medical assistants at the study sites to ensure the subjects’ safety and care. Additionally, the Ph.D. student will manage the overall administrative, technical, logistic, and fiscal responsibilities; communicate with all members of the research team; and will be responsible for the ethics application and renewal. The entire research team will work together on multivariate data analysis of processed spectra on SIMCA-P+ software, identifying nutritype signatures and determining their association with T2D.

### 2.7. Statistical Analysis

Statistical analysis will be performed using SPSS statistical software version 25.0 (IBM, Chicago, IL, USA). Data for categorical variables will be reported as number and percentage, while continuous data will be reported as mean ± standard deviation (SD). Data normality will be assessed prior to data analysis. The study will analyse the differences in the demographic characteristics between subjects who agreed to participate in the study versus those who did not agree to participate. Pearson’s correlation coefficient test will be used to determine correlations between continuous variables, the Chi-squared test to determine the association between categorical variables, and one-way analysis of variance (ANOVA) to compare the differences between the three groups. Multivariate linear regressions will determine the association between the metabolomic profile and factor scores of each dietary pattern yielded from PCA. Multinomial logistic regression will determine the factors, and the nutritype signature models associated with T2D. Variables significant at the bivariate level will be entered into the regression model as confounders. Crude and adjusted odds ratios (ORs) with a 95% confidence interval (CI) will be presented. Subgroup analyses will also be performed, such as early versus late postpartum (<1 year versus ≥1 year post-GDM), supplement versus nonsupplement users, and obese versus nonobese subjects. A previous study found different risk factors of T2D at early (<1 year) versus late postpartum (≥1 year) following GDM [4]. In case of missing data, the study will report the number of available data for the variable. We will analyse any differences in the characteristics between reporters versus nonreporters. A *p*-value < 0.05 for all statistical tests will be considered significant.

## 3. Discussion

The study aims to identify the nutritype signatures associated with T2D in women post-GDM using the ^1^H-NMR-based metabolomics approach. Cross-sectional studies in Malaysia found the that the prevalence of T2D in women post-GDM ranges between 0.8% and 6 weeks to 35.5% at 15 years postpartum [44,45,46]. This indicates that regardless of the year of diagnosis, having a previous diagnosis of GDM significantly increases future T2D risk.

Metabolomics has emerged as a crucial tool in discovering dietary biomarkers. Alterations in the metabolome are also implicated in disease development, as they respond to nutrients, disease and stress [19]. Traditional methods for assessing dietary intake, including FFQ, may be prone to bias, recall errors, and underreporting [47]. Dietary biomarkers, on the other hand, provide a more objective and accurate measure of food intake [47]. Together, the use of traditional dietary assessment and metabolomic profiles can improve the accuracy of dietary assessment [47].

In the Malaysian context, a six-month randomised controlled trial assigned 77 women post-GDM to either a low-glycaemic index (LGI) diet or a conventional healthy dietary recommendation [48]. The study reported that women in the LGI diet group lost significantly more weight and improved 2 h postprandial glucose levels. A few limitations of the use of the 3-day food record were underreporting by the subjects and the inability to capture the subjects’ habitual dietary patterns [48]. Hence, we propose the combination of a traditional dietary assessment together with a metabolomics approach to produce a more objective dietary assessment method. Additionally, dietary patterns of Malaysians may be different from those of the Caucasian population [10] and warrant further investigation in relation to T2D risk in this population.

Metabolomic profiles associated with dietary patterns have been previously studied in women post-GDM [14,15,16,17]. In the study by Andersson-Hall et al. involving 237 Scandinavian women post-GDM, linoleic acid was positively associated with vegetable oil and negatively associated with margarine intake. The linoleic acid level was reduced in women post-GDM who transitioned to T2D [14]. However, food intakes were calculated as the proportion of total diet (%) and not their absolute values in the study [14]. Another study involving 347 women post-GDM from the Nurses’ Health Study II found a correlation between dietary and plasma BCAAs [17]. Additionally, women post-GDM with higher dietary and plasma BCAAs had an elevated risk of T2D [17]. Serum and dietary polyunsaturated fatty acids were also correlated in the German study by Fugmann et al.; however, this study did not correlate dietary and serum fatty acids with the risk of T2D [15]. One study did not find any association between dietary glycaemic index and metabolites and did not assess the correlation between these parameters with the risk of T2D [16]. All of these studies [14,15,16,17] only focused on specific types of food or dietary components, instead of exploring dietary patterns as a whole. Only two studies [14,17] directly associated dietary intake and metabolites with the risk of T2D, indicating a paucity of evidence concerning nutritype signatures. Furthermore, all of these studies were conducted in the Western population, whose metabolomic profile may differ from that of the Asian population [49]. For instance, White European pregnant women had significantly lower glucose levels and most amino acid metabolites compared with South Asians, due to differences in body mass index and characteristics during pregnancy [49].

This study has a few limitations. The nature of the study is cross-sectional. Hence, the study will not be able to establish causation and effect. The study is limited to exploring only associations between dietary patterns, metabolomic profile, nutritype signatures, and T2D at a single time point. We also will not stratify subjects according to whether they have been practising certain treatments for T2D.

Our study aims to determine the associations between metabolomic profile and dietary patterns, and subsequently develop nutritype signature models. We will then determine which combination of dietary pattern and metabolomic profile is directly and significantly associated with T2D in women post-GDM. The findings on nutritype signatures will support the development of early prevention measures against T2D in women post-GDM. Metabolomics is commonly applied in nutrition research to assess responses to specific dietary patterns, discover biomarkers related to diet and disease, and subsequently determine the most effective dietary interventions to prevent diseases [50,51]. The field of nutritional metabolomics, which integrates metabolomics with nutrition in complex biosystems, may provide insight into the mechanisms linking diet and disease [48].

## Figures and Tables

**Figure 1 metabolites-12-00843-f001:**
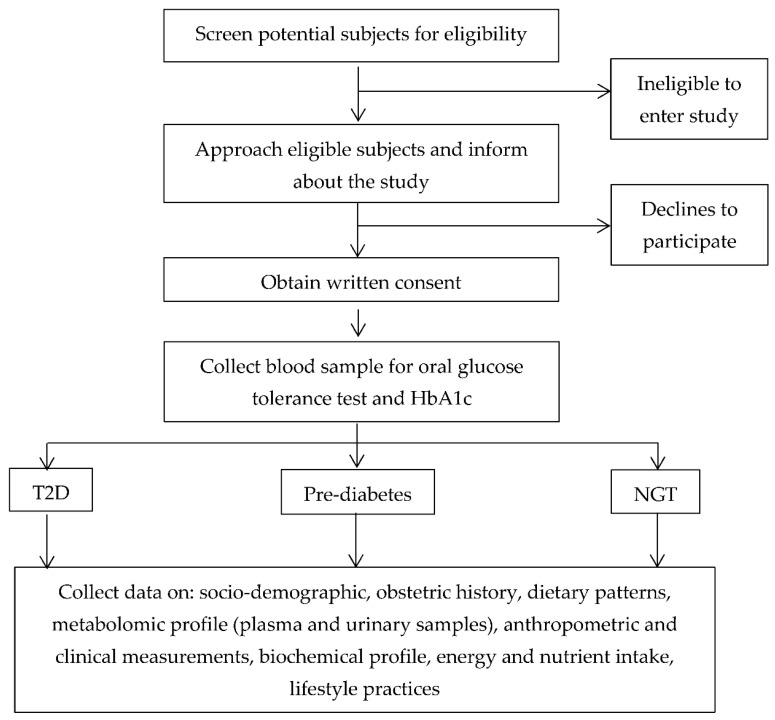
Research workflow.

**Table 1 metabolites-12-00843-t001:** Diagnostic values for normal glucose tolerance, type 2 diabetes, and prediabetes based on oral glucose tolerance test and HbA1c.

Category	Oral Glucose Tolerance Test		HbA1c (%)
	Fasting Plasma Glucose (mmol/L)	2-h Plasma Glucose (mmol/L)	
Normal glucose tolerance (NGT)	<6.1	<7.8	<5.7
Prediabetes, which includes: Impaired fasting glucose (IFG) Impaired glucose tolerance (IGT)	6.1–6.9	7.8–11.0	5.7–<6.3
Type 2 diabetes (T2D)	≥7.0	≥11.1	≥6.3

## Data Availability

Not applicable.

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
