# Peer review of "Dietary Patterns, Metabolomic Profile, and Nutritype Signatures Associated with Type 2 Diabetes in Women with Postgestational Diabetes Mellitus: MyNutritype Study Protocol"

_metabolites, 2022, doi:10.3390/metabo12090843_

Round 1

Reviewer 1 Report

Line 110

There is only 1 primary outcome, please define 1 singular outcome and the other (primary) outcomes to secondary outcomes.

Line 219

The results section suggests wrong expectations, as there can be no results in this publication yet. Maybe it is better to include this section to the methods section.

The manuscript is very logically structured and well written, and results are awaited with great excitement.

Author Response

Response to Reviewer 1

Point 1: Line 110

There is only 1 primary outcome, please define 1 singular outcome and the other (primary) outcomes to secondary outcomes.

Response 1: Amended 2.1 Study Design, lines 135-140:

The primary outcome is the nutritype signatures associated with T2D in Malaysian women post-GDM. The secondary outcomes of the study include the comparisons of anthropometric and clinical measurements, biochemical profile, energy and nutrient intake, and lifestyle practices between the three groups of women post-GDM (NGT, pre-diabetes and T2D).

Point 2: Line 219

The results section suggests wrong expectations, as there can be no results in this publication yet. Maybe it is better to include this section to the methods section.

Response 2: The reviewer’s comment is similar to the comment from the Academic Editor.

We have renamed the section to 2.5. Measures, and included it under Section 2. Methods.

Subheadings under this section are also renamed:

2.5.1. Diagnosis of type 2 diabetes and pre-diabetes

2.5.2. Sociodemographic characteristics and obstetric history

2.5.3. Dietary patterns

2.5.4. Metabolomic profile

2.5.5. Nutritype signatures

2.5.6. Anthropometric and clinical measurements

2.5.7. Biochemical profile

2.5.8. Energy and nutrient intake

2.5.9. Lifestyle practices

Point 3: The manuscript is very logically structured and well written, and results are awaited with great excitement.

Response 3: Thank you, we appreciate all comments from the reviewer.

Reviewer 2 Report

The protocol nicely describes the steps to study the nutritype signatures to predict the risk of Type 2 diabetes.

The manuscript does not address any specific question in the research of diabetes. It describes a protocol to study the nutritype signatures in individuals to predict the risk of Type 2 diabetes.

The topic is relevant. The methods, protocols, and control are appropriate.

The study describes only the protocol and once the results of this study are published then it may help to address the specific gap in the field.   

Author Response

Point 1: The protocol nicely describes the steps to study the nutritype signatures to predict the risk of Type 2 diabetes.

The manuscript does not address any specific question in the research of diabetes. It describes a protocol to study the nutritype signatures in individuals to predict the risk of Type 2 diabetes.

The topic is relevant. The methods, protocols, and control are appropriate.

The study describes only the protocol and once the results of this study are published then it may help to address the specific gap in the field.  

Response 1:

Thank you, we appreciate all comments from the reviewer.

Reviewer 3 Report

The review article by Hasbullah et al. describes the study to investigate the combined effects of dietary patterns and metabolomic Profile on predicting the risk of T2D in women post-GDM. However, this study will be completed by the end of 2022, and there are no results and figures related to the metabolomics of the plasma and urine samples of the participant of this study. I suggest after this study is completed, the authors reflect on the outcome of this study. Without any results, figures, and discussion, this paper should not be considered for publication.

Please consider the below points for your future submission:

Line 49 and 50. The authors mentioned “The multi-ethnic population of Malaysia is pre-dominantly made up of Malay, Chinese and Indian ethnic groups;

Why this diversity has made the country a perfect setting for the T2D epidemic”? Are there any references to support this statement?

Even though in the abstract you mentioned what does GDM acronym means, it is better to be repeated it in the introduction as “gestational diabetes mellitus (GDM)” when it is mentioned for the first time.

Line 143 to 166: Is it common to mention the task of each person in the section? Shouldn’t you put this information in “Author contribution” section?

Figure 1 where is the information about the NMR experiment. Why only blood sample is collected? You mentioned in the results section that NMR was performed on both plasma and urine samples.

Why the section 3.3. Metabolomic Profile is placed in the results section instead of the materials and methods section.

Author Response

Response to Reviewer 3

Point 1: The review article by Hasbullah et al. describes the study to investigate the combined effects of dietary patterns and metabolomic Profile on predicting the risk of T2D in women post-GDM. However, this study will be completed by the end of 2022, and there are no results and figures related to the metabolomics of the plasma and urine samples of the participant of this study. I suggest after this study is completed, the authors reflect on the outcome of this study. Without any results, figures, and discussion, this paper should not be considered for publication.

Response 1:

Thank you, we appreciate all comments from the reviewer and will reflect on the outcomes of the study once completed. We would like to clarify that this is not a review article, but a study protocol that outlines the design and research workflow of an ongoing study.

Point 2: Line 49 and 50. The authors mentioned “The multi-ethnic population of Malaysia is pre-dominantly made up of Malay, Chinese and Indian ethnic groups;

Why this diversity has made the country a perfect setting for the T2D epidemic”? Are there any references to support this statement?

Response 2: Lines 49-51 have been removed as the academic editor requests to shorten the introduction and focus more on the research gap. Hence, we removed the sentence as we feel it is not relevant to the study aims, which are about dietary patterns, metabolomic profile and nutritype signatures.

Point 3: Even though in the abstract you mentioned what does GDM acronym means, it is better to be repeated it in the introduction as “gestational diabetes mellitus (GDM)” when it is mentioned for the first time.

Response 3: Amended line 56: Women who had been previously diagnosed with gestational diabetes mellitus (post-GDM)…

Point 4: Line 143 to 166: Is it common to mention the task of each person in the section? Shouldn’t you put this information in “Author contribution” section?

Response 4: We want to elaborate on the tasks and responsibilities of each author. We moved the lines to Section 2.6. Governance and Ethics (as suggested by the academic editor), lines 388-410, to show how the study is being governed by the principal investigator and other study personnel. This method of writing is also shown in another study protocol already published in Metabolites journal (Vadset et al., 2022; doi.org/10.3390/metabo12010078).

Point 5: Figure 1 where is the information about the NMR experiment. Why only blood sample is collected? You mentioned in the results section that NMR was performed on both plasma and urine samples.

Response 5: Blood sample is collected only for OGTT and HbA1c. We added to Figure 1: collect data on metabolomic profile (plasma and urinary samples).

Point 6: Why the section 3.3. Metabolomic Profile is placed in the results section instead of the materials and methods section.

Response 6: The reviewer’s comment is similar to the academic editor’s comment. We have moved all data collection measurements to section 2.5 Measures (under 2. Methods).

Subheadings under this section are also renamed:

2.5.1. Diagnosis of type 2 diabetes and pre-diabetes

2.5.2. Sociodemographic characteristics and obstetric history

2.5.3. Dietary patterns

2.5.4. Metabolomic profile

2.5.5. Nutritype signatures

2.5.6. Anthropometric and clinical measurements

2.5.7. Biochemical profile

2.5.8. Energy and nutrient intake

2.5.9. Lifestyle practices

Round 2

Reviewer 3 Report

The authors have addressed all my concerns.

Author Response

Comment: The authors have addressed all my concerns.

Response:

Thank you, we appreciate all comments from the reviewer.
